# Dicarboxylic Acid-Based Co-Crystals of Pyridine Derivatives Involving Structure Guiding Unconventional Synthons: Experimental and Theoretical Studies

**Pranay Sharma** [1] , **Rosa M. Gomila** [2] , **Antonio Frontera** [2,*] , **Miquel Barcelo-Oliver** [2]
**and Manjit K. Bhattacharyya** [1,*]

[1] Department of Chemistry, Cotton University, Guwahati 781001, Assam, India
[2] Departament de Química, Universitat de les Illes Balears, Crta de Valldemossa km 7.7, 07122 Palma de Mallorca (Baleares), Spain
\* Correspondence: toni.frontera@uib.es (A.F.); manjit.bhattacharyya@cottonuniversity.ac.in (M.K.B.)

**Abstract:** Four co-crystals involving dicarboxylic acids and pyridine derivatives, viz. (ox)$_{0.5}$(2-CNpy) (**1**), (adp)(4-CNpy)$_2$ (**2**), (tp)(4-CNpy)$_2$ (**3**) and (adp)(3-CNpy)$_2$ (**4**) (ox = oxalic acid, tp = terephthalic acid, adp = adipic acid, CNpy = cyanopyridine), have been synthesized at room temperature in water medium. Crystal-structure analysis of co-crystal **1** reveals the presence of unconventional O···π(oxalic acid)-hole interaction with the C-C bond of *ox* moiety, along with parallel nitrile–nitrile interactions. The structural topologies of co-crystals **2**–**4** unfold the presence of antiparallel nitrile–nitrile interactions involving the *CNpy* moieties. The molecular associations involving the H-bonds and other unconventional contacts among the co-formers of the multicomponent co-crystals are analyzed using density functional theory (DFT) calculations combined with molecular electrostatic potential (MEP) surface, quantum theory of atoms-in-molecules (QTAIM) and noncovalent interaction (NCI) plot computational tools. The computational studies revealed the presence of unconventional O···π-hole interaction in **1** and the H-bonded synthons with π-stacked nitrile contacts involving *CNpy* moieties in co-crystals **2**–**4**. The energetic features of the noncovalent contacts reveal the crucial roles of the H-bonding synthons and π-stacking interactions in the multicomponent compounds.

**Keywords:** co-crystals; O···π-hole; nitrile–nitrile; H-bonding; DFT calculations

## 1. Introduction

Multicomponent compounds (co-crystals, molecular salts, polymorphs) have attracted the interest of researchers owing to their significant physicochemical and pharmaceutical properties [1–8]. Co-crystals are structurally homogenous crystalline compounds, which contain two or more neutral building blocks that are present in definite stoichiometric ratios [9]. In recent years, a great deal of research interest has been devoted to multicomponent crystallization in both the academic and industrial fields, specifically for drug-development purposes [10–13]. The main reason is their ability to modify essential physicochemical prosperities, viz. the solubility, stability and bioavailability of compounds through multicomponent co-crystallization, without altering their primary effects [14,15]. Apart from their pharmaceutical significance, co-crystals have found profound applications in sensing, as micro- and nanocrystalline co-crystals possess chemical reactivity that is promising for sensors [16,17].

Various noncovalent interactions such as hydrogen bonding, π-stacking, van der Waal's forces, etc., predominantly trigger the assembly of the multicomponent systems at a fixed stoichiometric ratio in ambient conditions [18–20]. Hydrogen bonding interactions are well-known for playing crucial roles in the molecular recognition, composition and stability of the co-crystals [21]. Hydrogen bonding interactions are often employed when designing a synthon, as they have been well-recognized as the most powerful force to

organize molecules in the solid state [22]. The functional groups of carboxylic acids, alcohols, amides and heterocyclic bases of co-crystallized conformers are readily involved in strong hydrogen bonding interactions [23,24]. The bioactivities of such compounds as well as the generation of supramolecular layered assemblies are mainly due to the presence of these functional groups [25]. The aromatic π-stacking interactions observed between the aromatic rings of cyclic ligands are imperative in the development of pharmaceutically active flexible materials [26]. π-stacking interactions are important for systems containing aromatic rings, which usually range from large biological systems to relatively small molecules [27,28]. However, recently explored noncovalent interactions, viz. σ- and π-hole contacts, have also significantly contributed to the stabilities of supramolecular architectures [29–31]. Noncovalent nitrile–nitrile interactions play a crucial role in the stabilization of the supramolecular assemblies and are, therefore, of particular interest from the crystal-engineering viewpoint [32].

A synthon can be defined as a constituent part of a molecule to be synthesized, which is considered as the basis of the synthetic routes [33]. Supramolecular synthon-based innovative synthetic pathways have been employed by various research groups to develop co-crystals with potential pharmaceutical applications [34–36]. Substituted pyridine and dicarboxylic acid based supramolecular synthons have also been explored to design and develop molecular solids with potential applications [37,38].

Pyridine is effectively employed in the pharmaceutical industries as a raw material for various drugs, vitamins and fungicides [39,40]. Substituted pyridine derivatives have already acquired a great position in the fields of medicinal chemistry and agrochemicals [41,42]. It has been well-established that co-crystals containing dicarboxylic acids are also pharmaceutically active [43,44]. Co-crystallization of pharmaceutically active co-formers can enhance the physiochemical properties of the compounds through the formation of intermolecular noncovalent interactions [12]. The knowledge of the noncovalent interactions of pyridine and related N-donor heterocyclic ligands with dicarboxylic acids may be effectively useful in pharmaceuticals and also in the synthesis of porous materials [45]. Therefore, a proper fusion of experimental and theoretical studies is of the utmost importance, to visualize the role of such interactions in co-crystallized compounds of dicarboxylic acids and N-donor heterocyclic compounds. However, various research groups have explored the supramolecular architectures of co-crystals and salts of pyridine derivatives with dicarboxylic acids [46,47]. Lakshmipathi et al. have recently reported three 4-(1-Napthylvinyl) pyridine-based co-crystals and explored the structure–mechanical-property correlation, which is relevant towards various applications for photo switches, mechanical actuators, etc. [48].

To explore the supramolecular assemblies involving substituted pyridines and dicarboxylic acids, herein, we have reported the synthesis and crystal structures of four multi-component co-crystals, viz. (ox)$_{0.5}$(2-CNpy) (**1**), (adp)(4-CNpy)$_2$ (**2**), (tp)(4-CNpy)$_2$ (**3**) and (adp)(3-CNpy)$_2$ (**4**). We have analyzed the pK$_a$ rule to categorize the synthesized multicomponent compounds as co-crystals duringthe self-assembly of the dicarboxylic acids and the pyridine derivatives. The nitrile fragments of the *CNpy* moieties of the co-crystals are involved in parallel and antiparallel nitrile–nitrile interactions, which stabilize the layered assembly of the crystal structures. The oxygen atom of *ox* moiety is involved in novel O⋯π-hole contacts with the C-C bond of neighboring *ox* moieties. We have further performed theoretical studies to analyze the H-bonding networks and π-stacking interactions along with the O⋯π-hole and nitrile–nitrile contacts observed in the solid state of the co-crystals, using various computational tools.

## 2. Materials and Methods

### 2.1. Materials and Methods

All the chemicals used in the present study, viz. oxalic acid, adipic acid, terephthalic acid, 4-cyanopyridine, 3-cyanopyridine and 2-cyanopyridine, were purchased from the

commercial sources and used as received. Perkin Elmer 2400 Series II CHNS/O analyzer was used to carry out the elemental analysis of the compounds.

### 2.2. Syntheses of the Co-Crystals

To synthesize the co-crystals, the respective dicarboxylic acids (ox (**1**), adp (**2**, **4**), tp (**3**)) and the cyanopyridine moieties (2-CNpy (**1**), 3-CNpy (**4**), 4-CNpy (**2**, **3**)) were dissolved in 10 mL of deionized water in 1:2 ratios in round-bottomed flasks and mechanically stirred at room temperature for about two hours (Scheme 1). The solutions obtained were kept in cooling environment (below 4 °C) for crystallization; from which colorless single crystals suitable for single-crystal X-ray diffraction were obtained after few days. Yields: 0.126 g (**1**, 84.5%), 0.294 g (**2**, 83.0%), 0.306 g (**3**, 81.1%), 0.301 g (**4**, 85%). Anal. calcd. for $C_{14}H_{10}N_4O_4$ (**1**): C, 56.38%; H, 3.38%; N, 18.78%; found: C, 56.32%; H, 3.29%; N, 18.65%; anal. calcd. for $C_{18}H_{18}N_4O_4$ (**2**): C, 61.01%; H, 5.12%; N, 15.81%; found: C, 60.91%; H, 5.09%; N, 15.75%; anal. calcd. for $C_{20}H_{14}N_4O_4$ (**3**): C, 64.17%; H, 3.77%; N, 14.97%; found: C, 64.11%; H, 3.69%; N, 14.91%; anal. calcd. for $C_{18}H_{18}N_4O_4$ (**4**): C, 61.01%; H, 5.12%; N, 15.81%; found: C, 60.95%; H, 5.07%; N, 15.70%.

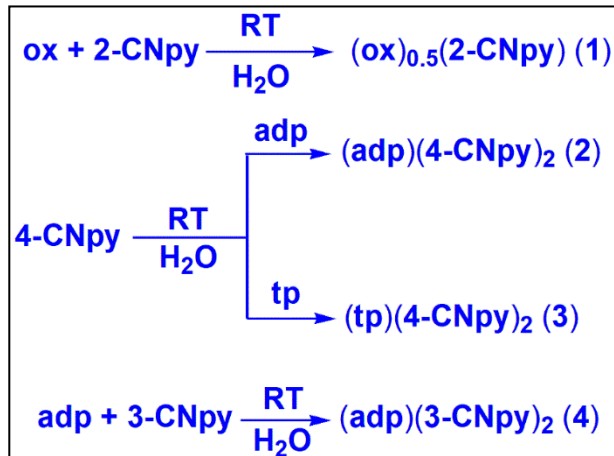

**Scheme 1.** Synthesis of compounds **1**–**4**.

### 2.3. Crystallographic Data Collection and Refinement

Crystal structures of co-crystals **1**–**4** were determined using single-crystal X-ray diffraction technique. X-ray diffraction data were collected using Bruker APEX-II CCD diffractometer with graphite monochromatized Cu/Kα radiation (λ = 1.54178 Å). Semi-empirical absorption correction as well as scaling and merging the different datasets for the wavelength were performed with SADABS [49]. The crystal structures were solved by direct method and refined by full matrix least-squares techniques with SHELXL-2018/3 [50] using the WinGX [51] software. Due to the relatively poor-quality crystal of compound **3**, it was moderately diffracting, and, therefore, some of the diffractions could not be included in the data integration and refinement, which results in comparatively higher R values [52]. The molecular structure and crystal packing of the compounds were drawn in Diamond 3.2. Collected data and refinement parameters for co-crystals **1**–**4** have been tabulated in Table 1.

CCDC 2122823, 2122838, 2122839 and 2122840 contain the supplementary crystallographic data for co-crystals **1**–**4**, respectively. These data can be obtained free of charge at http://www.ccdc.cam.ac.uk or from the Cambridge Crystallographic Data Centre, 12 Union Road, Cambridge CB2 1EZ, UK; fax: (+44) 1223-336-033; or email: deposit@ccdc.cam.ac.uk.

**Table 1.** Crystallographic data and structure refinement details for co-crystals **1–4**.

| Parameters | 1 | 2 | 3 | 4 |
|---|---|---|---|---|
| Formula | $C_{14}H_{10}N_4O_4$ | $C_{18}H_{18}N_4O_4$ | $C_{20}H_{14}N_4O_4$ | $C_{18}H_{18}N_4O_4$ |
| Formula weight | 149.13 | 354.36 | 374.35 | 354.36 |
| Temp (K) | 100.0 | 100.0 | 100.0 | 294 |
| Crystal system | Monoclinic | Triclinic | Triclinic | Triclinic |
| Space group | $P2_1/c$ | $P\overline{1}$ | $P\overline{1}$ | $P\overline{1}$ |
| a, (Å) | 3.664(2) | 3.8849(3) | 3.7596(4) | 4.0270(3) |
| b, (Å) | 13.887(8) | 10.5878(9) | 6.2540(7) | 10.8650(7) |
| c, (Å) | 13.455(6) | 11.7569(10) | 18.617(2) | 11.2177(7) |
| α, (°) | 90 | 63.760(4) | 92.569(4) | 67.878(2) |
| β, (°) | 91.55(2) | 86.193(5) | 95.589(4) | 86.725(2) |
| γ, (°) | 90 | 85.953(5) | 95.113(4) | 89.843(2) |
| Volume (Å³) | 684.5(6) | 432.34(6) | 433.29(8) | 453.84(5) |
| Z | 2 | 1 | 1 | 1 |
| Absorption coefficient (mm⁻¹) | 0.927 | 0.818 | 0.857 | 0.780 |
| F(000) | 308 | 186 | 194 | 186 |
| D(calcd) (Mg/m³) | 1.447 | 1.361 | 1.435 | 1.297 |
| Index ranges | $-4 \leq h \leq 3$, $-16 \leq k \leq 16$, $-16 \leq l \leq 16$ | $-3 \leq h \leq 4$, $-12 \leq k \leq 12$, $-13 \leq l \leq 14$ | $-4 \leq h \leq 4$, $-7 \leq k \leq 7$, $-22 \leq l \leq 22$ | $-4 \leq h \leq 4$, $-13 \leq k \leq 13$, $-13 \leq l \leq 13$ |
| Crystal size, (mm³) | $0.49 \times 0.20 \times 0.19$ | $0.45 \times 0.39 \times 0.33$ | $0.23 \times 0.15 \times 0.13$ | $0.28 \times 0.15 \times 0.12$ |
| θ range (°) | 6.374 to 66.541 | 4.195 to 67.223 | 4.776 to 133.11 | 8.524 to 140.024 |
| Independent reflections | 1114 | 1541 | 4640 | 1661 |
| Reflections collected | 1166 | 5081 | 4640 | 20138 |
| Refinement method | Full-matrix least squares on F² | Full-matrix least squares on F² | Full-matrix least squares on F² | Full-matrix least squares on F² |
| Data/restraints/parameters | 1166/0/105 | 1541/0/119 | 4640/0/129 | 1661/0/119 |
| Goodness-of-fit on F² | 1.098 | 1.063 | 1.094 | 1.089 |
| Final R indices (I > 2σ(I)) | R1 = 0.0438, wR2 = 0.1149 | R1 = 0.0479, wR2 = 0.1327 | R1 = 0.0761, wR2 = 0.2427 | R1 = 0.0453, wR2 = 0.1220 |
| R indices (all data) | R1 = 0.0452, wR2 = 0.1162 | R1 = 0.0539, wR2 = 0.1376 | R1 = 0.0785, wR2 = 0.2455 | R1 = 0.0468, wR2 = 0.1231 |
| Largest peak and hole (e·Å⁻³) | 0.22/−0.23 | 0.22/−0.24 | 0.43/−0.37 | 0.16/−0.15 |

## 2.4. Computational Methods

The energies and topological analyses of the supramolecular assemblies investigated herein were computed at the RI-BP86-D3/def2-TZVP [53,54] level of theory, using the crystallographic coordinates and the program Turbomole 7.2 [55], where only the positions of the H-atoms bonded to carbon were optimized. Grimme's D3 dispersion [53] correction has been used, since it is convenient for the correct evaluation of noncovalent interactions. This level of theory combining the BP86 functional, the def2-TZVP basis set and the D3 dispersion corrections has been previously used to analyze noncovalent interactions in the solid state, including those studied herein [56–59]. The QTAIM and NCI plot's [60] reduced density gradient (RGD) isosurfaces have been used to characterize noncovalent interactions, since both methods combined are useful to reveal noncovalent interactions in real space. The cubes needed to generate the NCI plot surfaces have been computed at the same level of theory using the wavefunctions generated by means of the Turbomole 7.2 program. The NCI plot index's isosurfaces correspond to both favorable and unfavorable interactions, as differentiated by the sign of the second density Hessian eigenvalue and defined by the isosurface color. The QTAIM analysis [61] and the cube files from the wavefunctions have been computed at the same level of theory by means of the MULTIWFN program [62] and are represented using VMD software [63]. The features of QTAIM analyses that we use here to characterize the H-bonds are bond paths and bond critical points (CPs). A bond path was defined by Bader as "a single line of maximum electron density linking the nuclei of two chemically bonded atoms" [61]. It begins at one nucleus and proceeds in the direction of maximum (most positive) gradient of the electron density, ρ, to a maximum, a bond critical point (BCP). The entire bond path runs from the first nucleus over the bond CP to the second nucleus.

## 3. Results

### 3.1. Syntheses and General Aspects

(ox)$_{0.5}$(2-CNpy) (**1**) has been synthesized by the reaction between one equivalent of *ox* and two equivalents of *2-CNpy* at room temperature in deionized water medium. Similarly, (adp)(4-CNpy)$_2$ (**2**) and (tp)(4-CNpy)$_2$ (**3**) have been synthesized by the reaction between two equivalents of *4-CNpy* with one equivalent of *adp* and *tp*, respectively. (adp)(3-CNpy)$_2$ (**4**) has been prepared by reacting one equivalent of *adp* and two equivalents of *3-CNpy* in water medium at room temperature (Scheme 1).

All the four compounds are fairly soluble in water and common organic solvents. It has been well-established that crystal-structure determination at low temperature can improve the quality of the datasets, resulting in more accurate structure determinations [64]. The crystal structures of co-crystals **1** and **2** have been redetermined at low temperatures with improved R and wR values (Tables S1 and S2), compared to those of the already-reported structures [65]. Improved R and wR values imply better experimental crystallographic data collection for compounds **1** and **2**, compared to the already reported structures [66,67]. We also have explored the presence of unconventional O···π-hole, parallel and antiparallel nitrile–nitrile interactions, directing supramolecular assemblies in compounds **1** and **2** (vide infra). Moreover; the energetic features of the supramolecular assemblies of the co-crystals have also been investigated using computational tools to gain an insight into the roles of the noncovalent contacts towards the solid-state stabilities of the compounds.

### 3.2. pK$_a$-Rule

One suitable approach to predict the formation of either co-crystal or molecular salt from two or more co-formers is pK$_a$-rule [68]. The ΔpK$_a$ (=pK$_a$ (base) − pK$_a$ (acid)) of two conformers can give possible indications whether a co-crystal or a salt will be formed [69–71]. A ΔpK$_a$ < 0 will result in a co-crystal, and a ΔpKa > 3 will lead to a salt; however, in the region between 0 < ΔpK$_a$ < 3, the predictability is elusive, as it may be co-crystal or salt formation that takes place [70–73]. Table 2 represents the pK$_a$ and ΔpK$_a$ values of the co-formers and multicomponent compounds of **1**–**4**. From the ΔpK$_a$ values, it is clear that all four compounds crystallize as co-crystals. All the dicarboxylic acids are associated with two pK$_a$ values (Table 2) corresponding to the presence of two carboxyl H-atoms. As soon as the first acidic proton of a dicarboxylic acid is deprotonated; it becomes an anion, from which it is harder to remove the proton than from the neutral dicarboxylic acid, resulting in a higher value for the second pK$_a$ than for the first [65].

**Table 2.** pK$_a$ and ΔpK$_a$ values of multicomponent crystals (**1**–**4**).

| Multicomponent Crystals | pK$_a$ (Base) | pK$_a$ (Acid) | pK$_a$ |
|---|---|---|---|
| (ox)$_{0.5}$(2-CNpy) (**1**) | −0.26 | 1.27/4.28 | −1.53/−4.54 |
| (adp)(4-CNpy)$_2$ (**2**) | 1.90 | 4.43/5.41 | −2.53/−3.51 |
| (tp)(4-CNpy)$_2$ (**3**) | 1.90 | 3.51/4.82 | −1.61/−2.92 |
| (adp)(3-CNpy)$_2$ (**4**) | 1.39 | 4.43/5.41 | −3.04/−4.02 |

### 3.3. Crystal Structure Analysis

The molecular structures of compounds **1**–**4** have been depicted in Figure 1a–d, respectively. (ox)$_{0.5}$(2-CNpy) (**1**) crystallizes in a monoclinic crystal system with a $P2_1/c$ space group. The asymmetric unit of **1** contains one molecule of *2-CNpy* and 0.5 molecules of *ox* moiety (Figure 1a). Crystal-structure analysis reveals the presence of center of inversion in **1** located at the mid-point of the C-C bond of *ox* moiety. (adp)(4-CNpy)$_2$ (**2**), (tp)(4-CNpy)$_2$ (**3**) and (adp)(3-CNpy)$_2$ (**4**) crystallize in a triclinic crystal system with a $P\bar{1}$ space group. The asymmetric units of **2** and **3** contain two molecules of *4-CNpy* and one molecule of *adp* and *tp*, respectively (Figure 1b,c). Co-crystal **4** contains one *adp* moiety and two *3-CNpy* moieties (Figure 1d) in the asymmetric unit. Further analysis reveals the presence of a

crystallographic center of the inversions in **2**, **3** and **4**, located at the mid-points of the C-C bonds in *adp* and at the centroid of the aromatic ring of *tp*, respectively.

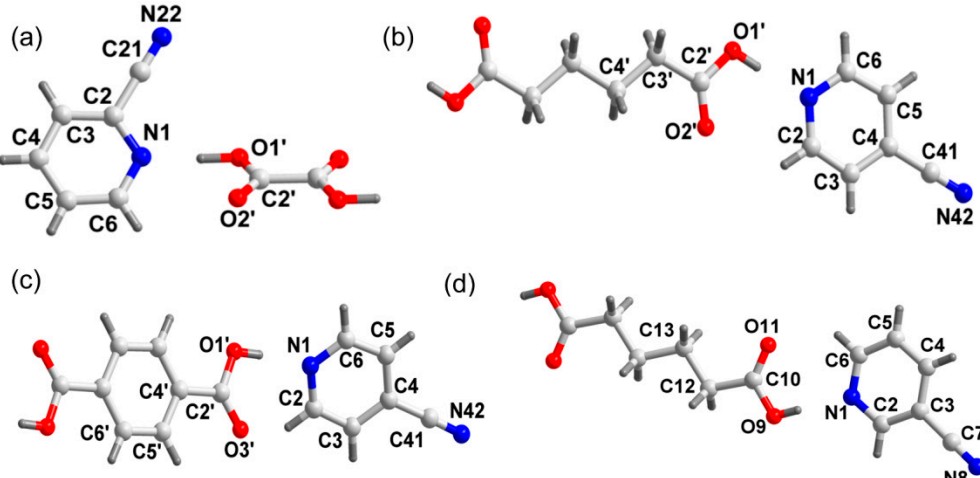

**Figure 1.** Molecular structures of (**a**) (ox)$_{0.5}$(2-CNpy) (**1**), (**b**) (adp)(4-CNpy)$_2$ (**2**), (**c**) (tp)(4-CNpy)$_2$ (**3**) and (**d**) (adp)(3-CNpy)$_2$ (**4**).

There is a simple method to determine if the pyridine ring of an organic moiety in multicomponent compounds is neutral or ionized. If the C–N–C bond angle of the pyridine ring is in the range of 117–118°, a neutral pyridine is indicated; meanwhile, an obtuse angle, in the range of 120–122°, indicates the ionization and formation of a pyridinium cation [74–77]. For compounds **1–4**, the C–N–C bond angles are around 117–118°, thereby suggesting the presence of co-crystals (see Table 3). X-ray crystallography can also provide evidence in support of the acid deprotonation of the carboxylate groups, by considering the C-O distances. The average bond lengths of C=O and C-O(H) are typically around 1.21 and 1.30 Å, respectively [78,79]. However, in carboxylate anions, the two C-O lengths are expected to be identical, about 1.25 Å [80,81]. In compounds **1–4**, the C=O and C-O(H) bond lengths are around 1.21 and 1.30 Å, respectively, which suggests the presence of neutral carboxylic acid groups in the co-crystals (see Table 4).

**Table 3.** C–N–C bond angles around the pyridine N-atoms in compounds **1–4**.

| Compound | C–N–C | Bond Angle (°) |
|---|---|---|
| Compound **1** | C2–N1–C6 | 117.4 |
| Compound **2** | C2–N1–C6 | 118.4 |
| Compound **3** | C2–N1–C6 | 117.9 |
| Compound **4** | C2–N1–C6 | 117.5 |

**Table 4.** C–O bond lengths of the carboxylic acid moieties of compounds **1–4**.

| Compound | C–O | Bond Length (Å) |
|---|---|---|
| Compound **1** | C2´–O1´ | 1.31 |
|  | C2´–O2´ | 1.21 |
| Compound **2** | C2´–O1´ | 1.33 |
|  | C2´–O2´ | 1.21 |
| Compound **3** | C2´–O3´ | 1.21 |
|  | C2´–O1´ | 1.32 |
| Compound **4** | C10–O11 | 1.19 |
|  | C10–O9 | 1.32 |

Crystal structure analysis of co-crystal **1** reveals the formation of two different supramolecular dimers involving the *ox* and *4-CNpy* moieties (Figure 2). The neighboring *ox* moieties of **1** are interconnected via unconventional O⋯π-hole interactions [82,83] to form a

supramolecular dimer (Figure 2a). The O1´ atom of an *ox* is involved in unconventional O···π-hole interactions with the π-electrons of the C-C bond of a neighboring *ox*, having a O1´···Cg distance of 3.14 Å. Such O···π-hole interactions involving the *ox* moieties are scarcely reported in the literature. Our research group has recently reported a similar O···π-hole interaction involving the C-C bond of the *ox* moieties in an oxalato bridged ternary Cu(II) complex, as first reported in [83]. Another interesting supramolecular dimer is also observed in the supramolecular assembly of **1**, involving the *4-CNpy* moieties (Figure 2b). The neighboring *4-CNpy* moieties are involved in unusual parallel nitrile–nitrile [84] and π-stacking interactions. Unconventional parallel nitrile–nitrile interactions are observed between the nitrile moieties of *4-CNpy* and the C···N distance of 3.66 Å. The C2–C21≡N22 angle is slightly deviated from linearity (179.6°), which may be due to the intermolecular supramolecular interactions involving the nitrile moieties [84]. Furthermore, aromatic π-stacking interaction is also observed in the supramolecular dimer having a centroid(N1, C2–C6)–centroid(N1, C2–C6) separation of 3.66 Å. These interactions have been further studied and characterized theoretically using various computational tools (vide infra).

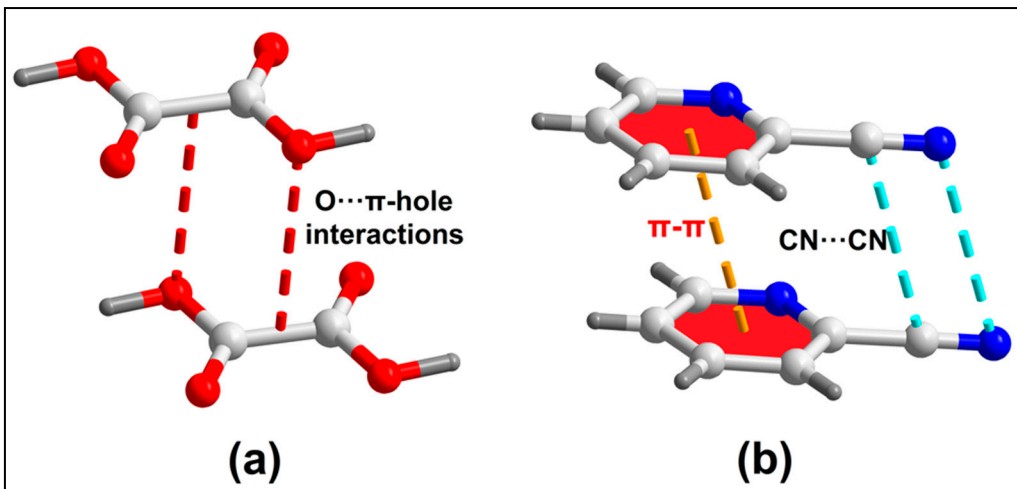

**Figure 2.** (**a**) Formation of supramolecular dimers of *ox* in **1**, assisted by unconventional O···π-hole interactions; (**b**) formation of supramolecular dimers of *4-CNpy* in **1**, assisted by unusual parallel nitrile–nitrile and π-stacking interactions.

The supramolecular dimers involving nitrile–nitrile contacts along with O–H···N and C–H···O hydrogen bonding interactions also stabilize the layered architecture of the compound along the crystallographic *ab* plane (Figure S1) (see Supplementary Materials). O–H···N hydrogen bonding interactions are observed in the layered assembly of **1**, involving the –OH moiety of *ox* and the N atom of the pyridine ring of *4-CNpy* having a O1´–H1···N1 distance of 1.61 Å (Table 5). Similarly, C–H···O hydrogen bonding interactions are observed involving the –CH moiety of *4-CNpy* and the O atom of the *ox* moiety having a C3–H3···O2´ distance of 2.77 Å.

The aforementioned supramolecular dimer (shown in Figure 2b) along with C–H···N hydrogen bonding interactions stabilize the layered assembly of the compound along the crystallographic *ac* plane (Figure S2). C–H···N hydrogen bonding interactions are observed involving the –CH moiety and the N atom of the *4-CNpy* moieties having a C5–H5···N22 distance of 2.99 Å.

**Table 5.** Selected parameters for hydrogen bonding interactions in **1–4**.

| D–H⋯A | d(D–H) | d(H⋯A) | d(D–A) | <(DHA) |
|---|---|---|---|---|
| **1** | | | | |
| C3–H3⋯O2´ | 0.95 | 2.77 | 3.358(2) | 120.9 |
| O1´–H1⋯N1 | 1.02 | 1.61 | 2.633(1) | 174.9 |
| C5–H5⋯N22 | 0.95 | 2.99 | 3.458(2) | 132.7 |
| **2** | | | | |
| C5–H5⋯N42 | 0.95 | 2.51 | 3.389(2) | 152.8 |
| C2–H2⋯O2´ | 0.95 | 2.66 | 3.331(1) | 127.4 |
| O1´–H1´⋯N1 | 0.84 | 1.83 | 2.668(1) | 172.6 |
| C4´–H4´B⋯N42 | 0.99 | 2.82 | 3.737(2) | 153.7 |
| **3** | | | | |
| C5–H5⋯N42 | 0.94 | 2.66 | 3.517(4) | 149.2 |
| N1–H1⋯O1 | 0.84 | 1.89 | 2.728(3) | 171.2 |
| C6–H6⋯O3´ | 0.95 | 2.28 | 3.124(2) | 145.6 |
| **4** | | | | |
| C4–H4⋯N8 | 0.93 | 2.58 | 3.442(2) | 154.1 |
| C12–H12B⋯N8 | 0.97 | 2.81 | 3.584(2) | 137.0 |
| C6–H6⋯O11 | 0.93 | 2.49 | 3.194(2) | 145.3 |
| C5–H5⋯O11 | 0.93 | 2.91 | 3.324(2) | 128.3 |
| C2–H2⋯O9 | 0.93 | 2.64 | 3.423(1) | 145.6 |
| O9–H9⋯N1 | 0.82 | 1.92 | 2.744(1) | 178.7 |
| C12–H12B⋯O9 | 0.95 | 2.97 | 3.654(3) | 133.6 |
| C13–H13A⋯O11 | 0.94 | 2.99 | 3.362(2) | 154.6 |

Crystal structure analysis of co-crystal **2** reveals the formation of a supramolecular dimer assisted by antiparallel nitrile–nitrile and C–H⋯N hydrogen bonding interactions (Figure 3). Antiparallel nitrile–nitrile interactions [85] are observed between the nitrile moieties of *4-CNpy* with aC⋯N separation of 3.47 Å. The C4–C41≡N42 angle is slightly deviated from linearity (179.3°), which may be due to the intermolecular supramolecular interactionsinvolving the nitrile moiety [86]. Moreover, C–H⋯N hydrogen bonding interactions are also observed in the supramolecular dimer having a C5–H5⋯N42 distance of 2.51 Å.

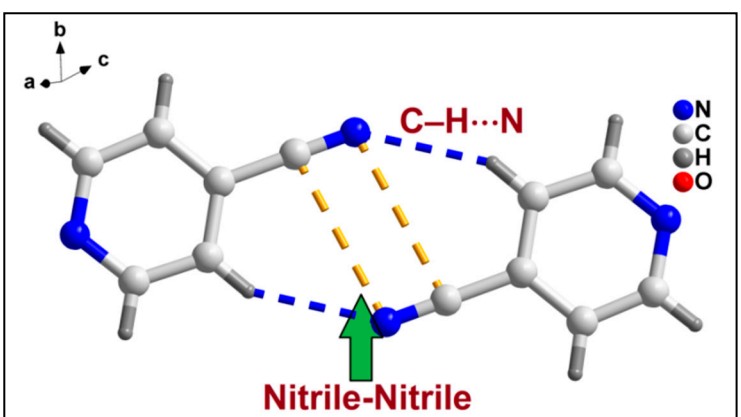

**Figure 3.** Formation of supramolecular dimer of co-crystal **2** involving *4-CNpy* moieties assisted by antiparallel nitrile–nitrile and C–H⋯N hydrogen bonding interactions.

Assemblies of these supramolecular dimers along with C–H⋯N, O–H⋯N and C–H⋯O hydrogen bonding interactions stabilize the layered architecture of co-crystal **2** along the crystallographic *bc* plane (Figure S3). The –CH moiety of *adp* is involved in C–H⋯N hydrogen bonding interactions with the N atom of *4-CNpy* having a C4´–H4´B⋯N42 distance of 2.82 Å. O–H⋯N hydrogen bonding interactions are also observed involving the –OH moiety of the *adp* ligand having a O1´–H1´⋯N1 distance of 1.93 Å. Moreover,

C–H···O hydrogen bonding interactions arealso present in the layered architecture having a C2–H2···O2´distance of 2.66 Å.

The aforementioned supramolecular dimers of co-crystal **2** (represented in Figure 3) also stabilize the layered assembly of the co-crystal along the crystallographic *ac* plane, assisted by aromatic π-stacking interaction (Figure S4). Parallel aromatic π-stacking interaction is observed in the crystal structure with a centroid(N1, C2-C6)–centroid(N1, C2–C6) separation of 3.88 Å.

Antiparallel nitrile–nitrile, aromatic π-stacking, C–H···N and O–H···N hydrogen bonding interactions are involved in the stabilization of the layered assembly of co-crystal **3** along the crystallographic *ac* plane (Figure S5). Antiparallel nitrile–nitrile interactions are observed involving the nitrile moieties of *4-CNpy* having aC···N separation of 3.77 Å. TheC4–C41≡N42 angle is found to be 178.4°; slightly deviated from linearity; this may be due to the intermolecular supramolecular interactions involving the nitrile moiety. Aromatic π-stacking interactions are observed involving the aromatic rings of the *tp* ligands having a centroid(C5´, C4´, C6´, C5´, C4´, C6´)–centroid(C5´, C4´, C6´, C5´, C4´, C6´) separation of 3.76 Å. Another π-stacking interaction involving the aromatic rings of *4-CNpy* having a centroid(N1, C2–C6)–centroid(N1, C2–C6) separation of 3.76 Å is observed in the crystal packing. C–H···N and O–H···N hydrogen bonding interactions are also observed in the crystal structure having C5–H5···N42 and O1´–H1´···N1 distances of 2.66 and 1.89 Å, respectively.

Figure 4 represents the layered assembly of co-crystal **3** along the crystallographic *bc* plane aided by antiparallel nitrile–nitrile and C–H···O hydrogen bonding interactions. Antiparallel nitrile–nitrile interactions are observed involving the nitrile moieties of *4-CNpy* having aC···N separation of 3.33 Å. Moreover, C–H···O hydrogen bonding interaction is also observed involving the O3´ atom of *tp* and the –CH moiety of *4-CNpy* (C6–H6···O3´ = 2.28 Å).

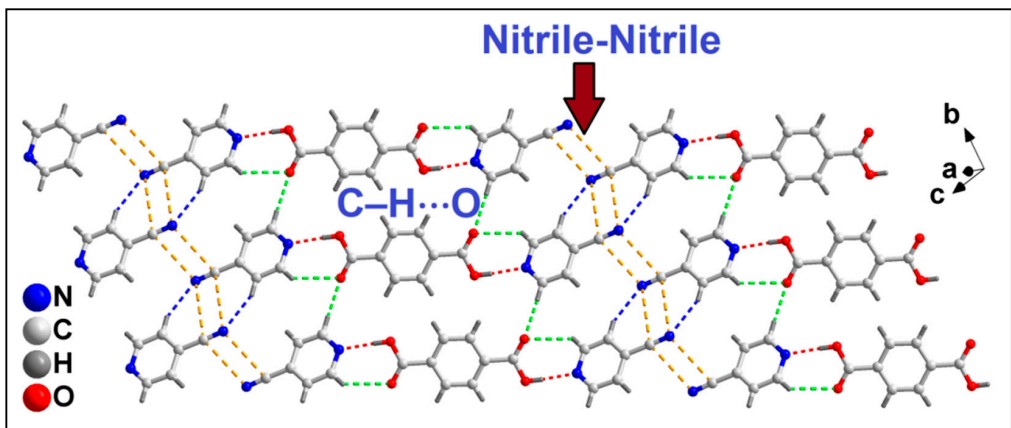

**Figure 4.** Layered assembly of co-crystal **3** along the crystallographic *bc* plane.

Crystal structure analysis of co-crystal **4** reveals the presence of antiparallel nitrile–nitrile, C–H···N, C–H···O and O–H···N hydrogen bonding interactions that stabilize the layered assembly of the compound along the crystallographic *bc* plane (Figure 5). The nitrile moieties of the adjacent *3-CNpy* ligands are separated by a distance of 3.58 Å, thereby involved in antiparallel nitrile–nitrile interactions. The corresponding C4–C41≡N42 angle is also deviated from linearity and is found to be 179.6°. C–H···N hydrogen bonding interactions are observed in the layered assembly of **4** having C4–H4···N8 and C12–H12B···N8 distances of 2.58 and 2.81 Å, respectively. Moreover, C–H···O hydrogen bonding interactions are also observed involving the O-atom of the *adp* and –CH moieties of *3-CNpy* having C6–H6···O11, C5–H5···O11 and C2–H2···O9 distances of 2.49, 2.91 and 2.64 Å, respectively. The –OH moiety of *adp* is involved in O–H···N hydrogen bonding interactions with the pyridine N atom of *3-CNpy* with aO9–H9···N1 distance of 1.92 Å.

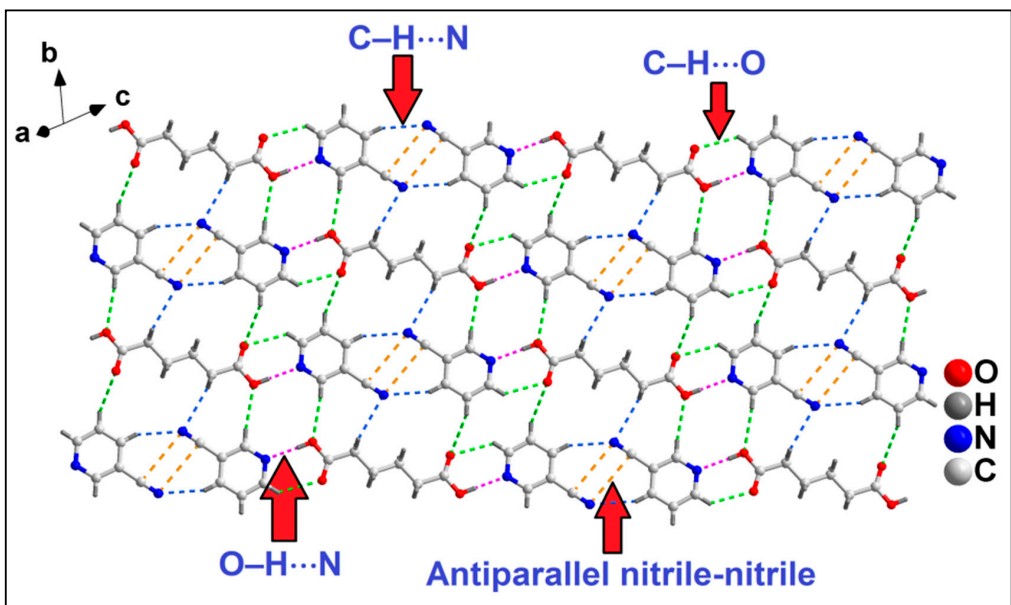

**Figure 5.** Layered assembly of co-crystal **4** along the crystallographic *bc* plane assisted by antiparallel nitrile–nitrile, C–H···N, C–H···O and O–H···N hydrogen bonding interactions.

Figure S6 depicts the layered assembly of co-crystal **4** along the crystallographic *ac* plane aided by aromatic π-stacking and C–H···O hydrogen bonding interactions. Aromatic π-stacking interaction is observed between the aromatic rings of *3-CNpy* moieties having a centroid(N1, C2–C6)–centroid(N1, C2–C6) separation of 3.92 Å, with the slipped angle of 19.2°. Moreover, C–H···O hydrogen bonding interactions are also observed involving the O-atoms and –CH moieties of neighboring *adp* moieties (C12–H12B···O9 = 2.97 Å; C13–H13A···O11 = 2.99 Å).

*3.4. Theoretical Study*

Initially, we have computed the molecular electrostatic potential (MEP) surfaces of the three *CNpy* moieties in order to compare their relative ability to participate in noncovalent interactions as electron donors (N-atoms) and acceptors (H-atoms). Figure 6 shows the MEP surfaces of *2-*, *3-* and *4-CNPy* molecules, evidencing that the nitrile N-atom is more nucleophilic than the pyridine. Moreover, the *2-CNPy* isomer is more nucleophilic than the rest. In contrast, all three isomers present similar MEP values at the H-atoms, thus suggesting a similar ability to participate in H-bonds as acceptors. Similarly, the MEP surfaces of the dicarboxylic acid used in this work have been also computed (see Figure 7), evidencing that the *ox* is the most acidic, followed by the *tpH₂* and *adp*, in line with their pK$_a$ values (1.27, 3.51 and 4.43, respectively). Their ability as H-bond acceptor (via the carbonyl O-atom) is similar in all acids. It is worth commenting that the MEP value is positive over the middle of the C–C bond in *ox*, thus providing an explanation to the O···π-hole interactions described above.

Co-crystals **2**–**4** present similar assemblies in the solid state, where the cyanopyridine rings form centrosymmetric dimers held together by C–H···N and antiparallel CN···CN interactions. Moreover, these dimers interact with the dicarboxylic acids, forming strong O–H···N and weak C–H···O H-bonds. Those assemblies have been analyzed, and the results gathered in Figure 8, including the QTAIM/NCI plot analyses. It can be observed that the three assemblies have similar dissociation energies (~21 kcal/mol). Such large dissociation energy confirms the relevance of these concurrent motifs in the solid state of the co-crystals. The QTAIM shows that each H-bond is characterized by a bond critical point (CP, represented by a red sphere in Figure 8) and bond path (represented as an orange line) connecting the H to the N-atoms of nitrile or pyridine or the O-atoms of the carboxylic groups. The interactions are also revealed by the NCI plot index, showing blue reduced

density gradient (RDG) isosurfaces for the O–H⋯N H-bonds and green RDG isosurfaces for the C–H⋯O/N bonds, thus evidencing the strong and weak nature of the H-bonds, respectively. This is further confirmed by the dissociation energies computed for each H-bond contact (values in red in Figure 8, next to the bond CPs).

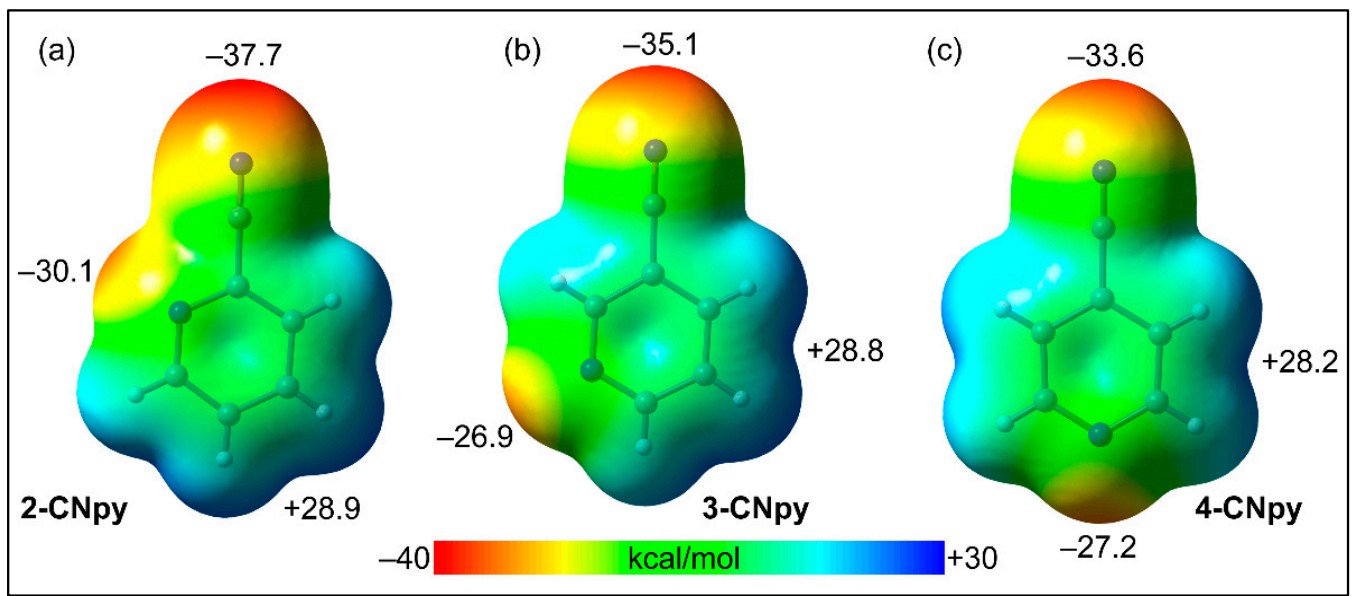

**Figure 6.** MEP surfaces of compounds *2-CNPy* (**a**), *3-CNPy* (**b**) and *4-CNPy* (**c**) at the PB86-D3/def2-TZVP level of theory. The energies at selected points are given in kcal/mol. Isosurface 0.001 a.u.

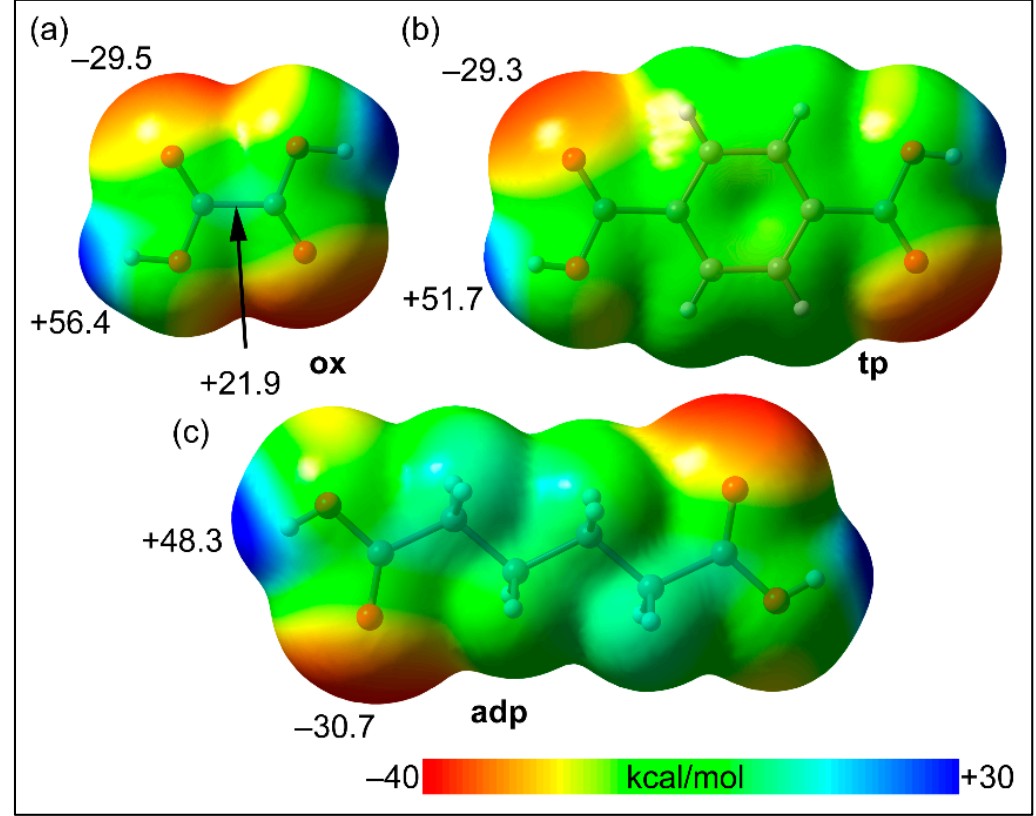

**Figure 7.** MEP surfaces of dicarboxylic acids; *ox* (**a**), *tp* (**b**) and *adp* (**c**) at the PB86-D3/def2-TZVP level of theory. The energies at selected points are given in kcal/mol. Isosurface 0.001 a.u.

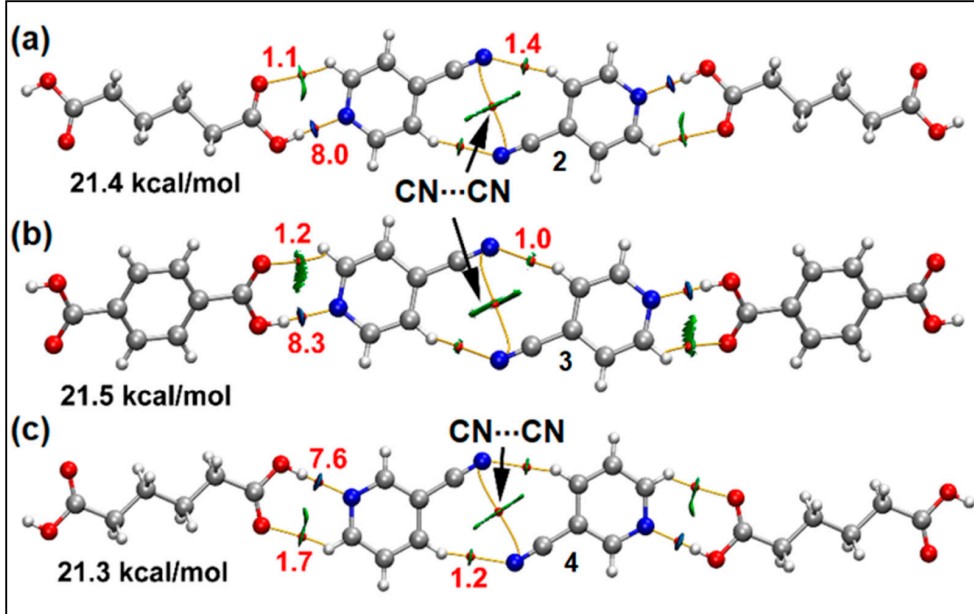

**Figure 8.** QTAIM and NCI plot analysis of the H-bonded assemblies of compounds **2**, **3** and **4**. The QTAIM energies are given in red for the H-bonds (in kcal/mol), and values in black correspond to the sum all the energies of all contacts.

These values have been computed using the $V_r$ energy predictor [87]. The O–H···N bond dissociation energies range from 7.6 kcal/mol in **4** to 8.3 kcal/mol in **3**, and the CH···O/N H-bonds range from 1.0 to 1.7 kcal/mol. It is also worth mentioning that the existence of the antiparallel CN···CN interaction is confirmed by both the QTAIM and NCI plot methods, showing a bond CP interconnecting the N-atoms of both cyano groups and a green isosurface between both groups. The energies associated to these contacts are very small, ranging from 0.3 kcal/mol in **4** to 0.5 kcal/mol in **3**. In comparison with the H-bond energy of the water dimer (–3.24 kcal/mol) [88], the OH···N H-bonds are significantly stronger, and the CH···O/N bonds are weaker.

For compound **1**, we have analyzed a similar assembly, where *ox* interacts with two *2-CNPy* rings (Figure 9a). In this case, the H-bond dissociation energy is very large (12.5 kcal/mol), in line with the strong acidity of the oxalic acid. The NCI plot analysis disclosed a green RDG isosurface between the O-atom of *ox* and one aromatic proton of the *2-CNPy* ring, thus suggesting a small C–H···O contribution to the formation of this trimeric assembly. We have also analyzed two additional interactions that are also important in the solid state, as described above (see Figure 2). For the π-stacked dimer, the QTAIM shows three bond CPs and bond paths connecting only the pyridine rings. However, the NCI plot analysis shows that the green RDG isosurface embraces the π-systems of both the pyridine and the cyano groups, thus suggesting the participation of the CN groups. In fact, the dimerization energy (–3.2 kcal/mol) is slightly greater in absolute value than that reported for the benzene dimer (–2.7 kcal/mol) [89], thus suggesting that the cyano groups have a favorable influence on the π-stacking.

Regarding the oxalic dimer (see Figure 9c), the dimerization energy is smaller than that of *2-CNPy* dimer. The QTAIM shows the two CPs and bond paths connecting the O-atoms to the C-atoms, thus confirming the π-hole nature of these contacts. The NCI plot isosurface embraces the whole π-system, thus suggesting that this interaction can be also envisaged as a π-stacking.

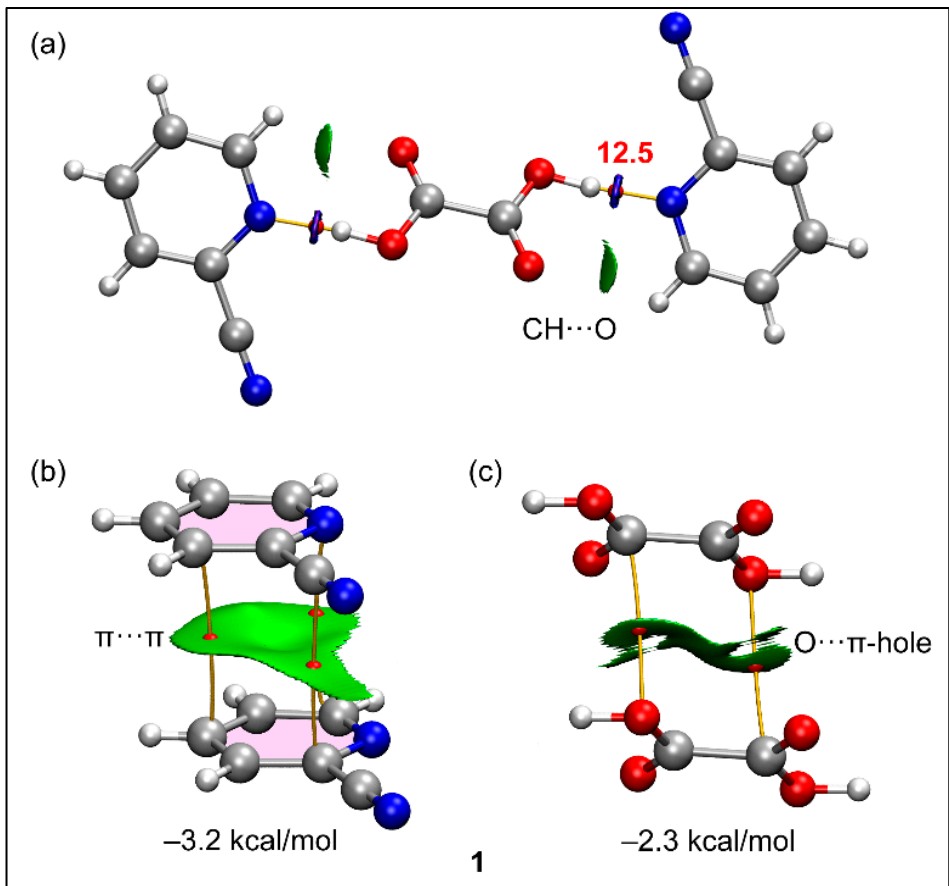

**Figure 9.** QTAIM and NCI plot analysis of the (**a**) H-bonded, (**b**) π-stacking and (**c**) π-hole assemblies of compound **1**. The QTAIM energies are given in red for the H-bonds (in kcal/mol).

## 4. Conclusions

Four co-crystals involving dicarboxylic acids and pyridine derivatives have been synthesized and characterized using the single-crystal X-ray diffraction technique. Crystal structure analysis of co-crystal **1** reveals the presence of unconventional parallel nitrile–nitrile interactions involving the nitrile moieties of *2-CNpy*. Further analysis reveals the involvement of the O-atom of *ox* in unconventional O···π-hole interactions with the π-hole located on the C-C bond of neighboring *ox* moiety. Crystal structure analysis of co-crystals **2**, **3** and **4** unfold the presence of a structure guiding the antiparallel nitrile–nitrile synthons involving the nitrile moieties of *CNPy*, which provide stabilities to the layered architectures of the co-crystals. Various H-bonding interactions also provide additional reinforcement to the crystal structures. We have further investigated the energetic features of the supramolecular assemblies observed in the crystal structures using DFT calculations, MEP surface, QTAIM and NCI plot computational tools. The strongest H-bonding interactions correspond to the O–H···N and O–H···O with dissociation energies that range from 12.6 to 17.8 kcal/mol. The π–π and O···π-hole interactions are significantly weaker (–3.2 and –2.3 kcal/mol, respectively). The energetic aspects of the supramolecular assemblies of the compounds reveal the significance of H-bonding and π-stacking interactions in governing the solid-state stabilities of the compounds.

**Supplementary Materials:** The following supporting information can be downloaded at: https://www.mdpi.com/article/10.3390/cryst12101442/s1. Figures S1–S6 and Tables S1 and S2 (see supplementary materials).

**Author Contributions:** Conceptualization, A.F. and M.K.B.; methodology, A.F.; software, A.F. and R.M.G.; formal analysis, A.F.; investigation, P.S. and R.M.G.; data curation, M.B.-O.; writing—original

draft preparation, P.S. and M.K.B.; writing—review and editing, M.K.B.; visualization, A.F.; supervision, M.K.B.; project administration, A.F.; funding acquisition, A.F. All authors have read and agreed to the published version of the manuscript.

**Funding:** Financial support from ASTEC, DST, the Government of Assam (grant number: ASTEC/S&T/192(177)/2020-2021/43) and the Gobierno de España, MICIU/AEI (projects numbers. EQC2018-004265-P and PID2020-115637GB-I00) are gratefully acknowledged.

**Data Availability Statement:** Not applicable.

**Conflicts of Interest:** The authors declare no conflict of interest. The funders had no role in the design of the study; in the collection, analyses or interpretation of data; in the writing of the manuscript; or in the decision to publish the results.

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
