# Peer review of "Dicarboxylic Acid-Based Co-Crystals of Pyridine Derivatives Involving Structure Guiding Unconventional Synthons: Experimental and Theoretical Studies"

_crystals, doi:10.3390/cryst12101442_

Round 1
Reviewer 1 Report
In the manuscript "Dicarboxylic Acid-based Co-crystals of Pyridine Derivatives Involving Structure Guiding Unconventional Synthons: Experimental and Theoretical Studies" the authors reported the synthesis, crystallographic and computational study of four co-crystals involving dicarboxylic acids and pyridine derivatives. The authors performed a detailed analysis of non-covalent bonding patterns using the geometrical data from crystal structures. Non-covalent bonding was also studied by DFT calculations combined with the molecular electrostatic potential surface, QTAIM theory, and NCI plots. In my opinion, the results of the study are very useful, and submited manuscript should be accepted for publication after addressing some minor issues:
1) It is not clear why the authors used the RI-BP86-D3/def2-TZVP level of theory for DFT calculations. I believe that this level of theory is appropriate, but a brief explanation would be very useful.
2) Authors calculated energies of different non-covalent interactions present in studied structures. I think that it would be very useful if the authors would compare obtained interaction energies with previously calculated interaction energies of hydrogen bonds in some well-known systems (like interaction energies between two water molecules for hydrogen bonds or interactions between two benzene molecules for pi-pi interactions). This type of comparison would point out the significance of the obtained results.
Author Response
First, we would like to thank this referee for his/her careful reading of the manuscript, corrections and suggestions. We have revised the manuscript accordingly. The point-by-point responses follow:
Comment 1: It is not clear why the authors used the RI-BP86-D3/def2-TZVP level of theory for DFT calculations. I believe that this level of theory is appropriate, but a brief explanation would be very useful.
Reply: We have added a brief explanation in the computational method section justifying the level of theory
Comment 2: Authors calculated energies of different non-covalent interactions present in studied structures. I think that it would be very useful if the authors would compare obtained interaction energies with previously calculated interaction energies of hydrogen bonds in some well-known systems (like interaction energies between two water molecules for hydrogen bonds or interactions between two benzene molecules for pi-pi interactions). This type of comparison would point out the significance of the obtained results.
Reply: Thank you for this suggestion. This has been done in the theoretical section of the results and discussion section.
Reviewer 2 Report
This paper gives a detailed study of four multi-component co-crystals experimentally and theoretically. Overall, the paper was organized with reasonable clarity. I think the manuscript can be published in “Crystals” after minor revision. The detailed comments are as follows:
(1) The last paragraph in Section “1.Introduction” aiming to give a brief introduction of this work is too long. The authors need to rephrase clearly.
(2) The synthetic methods of the four cocrystals (lines 107-135) are similar, which can be organized in one paragraph, instead of repetitive elaboration.
(3) “After a few days, colorless block shaped suitable single crystals were obtained from the slow evaporation of the mother liquor.” (line 112). What is the experimental condition of “the slow evaporation of the mother liquor”? As the crystallization of the cocrystals were in cooling conditions (2-4°C), the evaporation of water is difficult in this condition.
(4) In Table 2, the pKa(acid) has two values, what is the meaning of this region? It should be explained if possible.
Author Response
First, we would like to thank this referee for his/her careful reading of the manuscript, corrections and suggestions. We have revised the manuscript accordingly. The point-by-point responses follow:
Comment 1: The last paragraph in Section “1.Introduction” aiming to give a brief introduction of this work is too long. The authors need to rephrase clearly.
Our response: We thank the referee for his/her kind comments. We have now rephrased the Introduction section as suggested.
Comment 2: The synthetic methods of the four cocrystals (lines 107-135) are similar, which can be organized in one paragraph, instead of repetitive elaboration.
Our response: We thank the referee for his/her kind suggestion. We have now reorganized the synthesis in one paragraph as suggested.
Comment 3: “After a few days, colorless block shaped suitable single crystals were obtained from the slow evaporation of the mother liquor.” (line 112). What is the experimental condition of “the slow evaporation of the mother liquor”? As the crystallization of the cocrystals were in cooling conditions (2-4°C), the evaporation of water is difficult in this condition.
Our response: We thank the referee for his/her kind comment. We have now revised the sentence for clarity of the readers.
Comment 4: In Table 2, the pKa(acid) has two values, what is the meaning of this region? It should be explained if possible.
Our response: We thank the referee for his/her kind comment. We have now explained this part in the revised manuscript as suggested by the reviewer.